# Polysaccharides from Medicine and Food Homology Materials: A Review on Their Extraction, Purification, Structure, and Biological Activities

**DOI:** 10.3390/molecules27103215

**Published:** 2022-05-17

**Authors:** Jiaqi Xu, Jinling Zhang, Yumei Sang, Yaning Wei, Xingyue Chen, Yuanxin Wang, Hongkun Xue

**Affiliations:** 1College of Clinical Medicine, Hebei University, No. 342 Yuhua East Road, Lianchi District, Baoding 071002, China; xujiaqi1386@163.com; 2College of Traditional Chinese Medicine, Hebei University, No. 342 Yuhua East Road, Lianchi District, Baoding 071002, China; z15133838423@163.com (J.Z.); kq958649362@163.com (Y.S.); w17325289795@163.com (Y.W.); cxy18330531661@163.com (X.C.); sylvia202204@163.com (Y.W.); 3Key Laboratory of Particle & Radiation Imaging, Ministry of Education, Department of Engineering Physics, Tsinghua University, No. 30 Shuangqing Road, Haidian District, Beijing 100084, China

**Keywords:** polysaccharides, medicine and food homology, extraction and purification, structure, biological activities

## Abstract

Medicine and food homology (MFH) materials are rich in polysaccharides, proteins, fats, vitamins, and other components. Hence, they have good medical and nutritional values. Polysaccharides are identified as one of the pivotal bioactive constituents of MFH materials. Accumulating evidence has revealed that MFH polysaccharides (MFHPs) have a variety of biological activities, such as antioxidant, immunomodulatory, anti-tumor, hepatoprotective, anti-aging, anti-inflammatory, and radioprotective activities. Consequently, the research progress and future prospects of MFHPs must be systematically reviewed to promote their better understanding. This paper reviewed the extraction and purification methods, structure, biological activities, and potential molecular mechanisms of MFHPs. This review may provide some valuable insights for further research regarding MFHPs.

## 1. Introduction

The theory of homology between food and medicine first appeared in the Yellow Emperor’s Internal Classic. Later, Shennong’s herbal classic and Qianjin recipe improved this theory to varying degrees. In the traditional Chinese medicine industry, people are used to calling the substances that are both food and traditional Chinese medicine “medicine food homology (MFH) materials”. Recently, with the improvement in people’s health awareness, medical thought has gradually changed from an initial treatment to the combination of prevention and treatment. Therefore, the natural products of MFH materials have attracted extensive attention. Currently, 110 kinds of MFH materials are listed according to the China’s National Health Commission and State Administration of Market Supervision [1]. However, few Chinese herbal medicines are developed and applied. Medicine and food homology (MFH) materials are rich in polysaccharides, proteins, fats, vitamins, and other components [2]. Recently, several bioactive components of MFH materials have attracted increasing attention with the rise of the concept of healthy diet. Polysaccharides, as one of the most important active components in MFH materials, are long-chain carbohydrates formed by complex polymerization of aldose or ketose through glycosidic bonds [3,4]. MFH polysaccharides (MFHPs) have various biological activities such as hypoglycemic [5], hypolipidemic [6], antioxidant [7], and immune regulation [8] properties owing to their special structure, non-toxic nature and lack of side effects [9,10]. However, due to the different structural characteristics of MFHPs separated by different extraction and purification techniques, the bioactivities also have a certain difference [4]. Hence, the extraction, purification, structure, and chemical modification of MFHPs have been widely investigated, and a certain research foundation has been achieved.

A high molecular weight leads to high viscosity of polysaccharides [11]. High viscosity has become a major problem in the research of different fields of polysaccharides, and the difficulty of production greatly increases their cost, resulting in the challenging large-scale application of polysaccharides [12]. Moreover, the literature about the exact structural information of MFHPs is limited. According to the existing literature, MFHPs could be used as functional active ingredients and therapeutic agents for disease prevention and treatment. Hence, the research progress and future prospects of MFHPs must be systematically reviewed to better understand MFHPs. In this paper, studies on the extraction and purification methods, structure, and biological activities of MFHPs in recent years are summarized (Figure 1), and the research progress and future development trend are described in detail. These findings provide an important scientific foundation for the development and use of MFHPs in the future.

## 2. Extraction and Purification Methods for MFHPs

Effective extraction and purification for MFHPs are the primary premise behind studying the structure and biological activities of polysaccharides. With growing attention being paid to the medicinal value and health benefits of MFHPs, a series of conventional and novel extraction methods have already been developed. Figure 1 shows the extraction and purification process of MFHPs.

MFHPs require efficient extraction without destroying the essential properties of polysaccharides. Polysaccharide is a kind of polar macromolecule compound, which exists in animal cell membrane, plant cell wall and microbial cell wall [13]. As one of the four fundamental components of life, polysaccharides play a crucial role in illness prevention and therapy [14,15]. The classical hot water extraction (HWE) method is widely used in the extraction of polysaccharides because of its low extraction cost and ease of use. Based on a thorough review of previous literature, it was found that MFHPs could normally be extracted by HWE method under the following circumstances: Extraction temperature of 60–100 °C, liquid-to-solid ratio of 5:1–30:1 mL/g, and extraction time of 1–4 h [16,17,18,19,20]. Besides HWE method, traditional extraction methods include acid–base extraction (ACE) and alkaline–base extraction (ALE) methods. These extraction methods could shorten the extraction time to a certain extent, whereas the acidic and alkaline conditions may lead to the cleavage of glycosidic bonds and destroy the structure of polysaccharides [21]. In summary, although the traditional extraction methods have certain advantages, they have significant disadvantages, such as being time-consuming, and having low efficiency, large solvent consumption and high temperature [21,22]. Hence, the application of these extraction methods is limited.

To improve the extraction efficiency of polysaccharides, some advanced and effective extraction methods need to be further developed based on the tenet of enhancing cell wall decomposition without harming the structure of MFHPs. Notably, enzyme-assisted extraction (EAE), microwave-assisted extraction (MAE), and ultrasound-assisted extraction (UAE) methods have been widely used in the extraction of MFHPs. EAE is an advanced extraction method due to its high efficiency and environmental friendliness through the use of enzymes to destroy the cell wall and reduce the mass transfer resistance of polysaccharides [23]. This method has mild extraction conditions and is less destructive to the polysaccharide’s structure. The main enzymes currently used in EAE are pectinase, cellulase, and papain. Yu et al. prepared complex enzymes (pectinase, cellulase, and papain in the ratio of 2.5:1.7:2.1, g/g), and then added citric acid-disodium hydrogen phosphate buffer (pH 4.0) at a liquid to material ratio of 10:1 mL/g [24]. The enzymatic reaction was carried out in an oscillating bath (55 °C) for 2.6 h to obtain Berry’s water-soluble polysaccharides [24]. The yield of polysaccharides (4.09%) obtained by EAE was higher than that of HWE (3.44%). The anti-tumor activity of Ginger polysaccharide fraction (EGP2) obtained by complex enzyme-assisted extraction against MCF-7 was significantly higher than that of Ginger polysaccharides obtained by hot water extraction (HGP) [19]. MAE is a potential extraction method for polysaccharides from natural resources. It is a technique and method for extracting various chemical components from natural plants, minerals or animal tissues in a microwave reactor with a suitable solvent. The principle is that heating causes water to evaporate in the target cells, thereby generating tremendous pressure to cause the cells to rupture, to achieve the purpose of distributing the contents [25]. Wu et al. obtained polysaccharides from Cassia seed (CSP-M) by MAE and the optimal extraction process by response surface methodology [26]. The optimal combination of extraction process parameters is as follows: Microwave power of 415 W, extraction time of 7.0 min, and liquid-to-material ratio of 51 mL/g [26]. CSP-M displayed good antioxidant activity, which can be related to its low molecular weight and high amount of unmethylated galacturonic acid [26]. Li et al. used single-factor experiments and orthogonal optimization experiments to optimize the extraction process of polysaccharides [27]. The optimal extraction process to achieve the highest yield of polysaccharides (7.82%) from *Angelica sinensis* was obtained at the microwave power of 500 W, extraction time of 20 min and liquid-to-solid ratio of 15 mL/g [27]. UAE utilizes the cavitation effect and powerful shear force generated by ultrasound to accelerate mass transfer and reduce the extraction time and extraction temperature, making it a potential extraction method for thermally unstable compounds such as polysaccharides [28]. Hu et al. found that the maximum extraction rate (8.340%) was obtained at ultrasonic power of 480 W, ultrasonic time of 16 min, and liquid to material ratio of 21 mL/g via response surface methodology [29]. Compared with HWE, UAE has higher extraction efficiency and lower molecular weights of polysaccharides. Guo et al. found that the optimum extraction parameters with the highest polysaccharide yield (10.29%) were obtained at an ultrasonic power of 144 W, extraction time of 180 min, liquid to material ratio of 60 mL/g, and extraction temperature of 80 °C [30]. Song et al. investigated the effect of extraction temperature, extraction time, material-to-liquid ratio, and ultrasonic power on the extraction rate of polysaccharides from *Lycium barbarum* (LBP) and optimized the extraction process by response surface methodology [31]. The results indicate that the optimal extraction process to achieve the highest LBP yield (12.54 ± 0.12)% was obtained at the ultrasonic power of 185 W, extraction time of 80 min, extraction temperature of 73 °C, and liquid-to-solid ratio of 38 mL/g [31]. Table 1 summarizes the current extraction methods of MFHPs. Currently, a combination of various extraction technologies is used for the extraction of polysaccharides to achieve their efficient extraction and separation. Then, the crude polysaccharides extracts are dialyzed, precipitated, and freeze-dried to obtain purified MFHPs.

Due to the limitations of extraction methods, the crude polysaccharides contain various impurities, such as pigment, protein, monosaccharide, and other substances. Hence, the isolation and purification of crude MFHPs are extremely important to obtain homogenous polysaccharides fractions, identify their structural features, and determine biological activities. Many kinds of purification methods are used to further purify crude polysaccharides. According to the purification mechanism and process, purification methods of polysaccharides can be divided into three categories: (1) Physical separation based on differences in molecular weight and solubility; (2) Column chromatographic separation and purification method based on different intermolecular forces; (3) the chemical precipitation method based on different solubility in solvent. Each purification method has its own advantages and disadvantages. Hence, the application scope of different separation methods is also different. Table 2 summarizes the various polysaccharide purification techniques available. Traditionally, the crude MFHPs are deproteinized and decolorized using the Sevage method and hydrogen peroxide treatment method, respectively. Subsequently, the crude MFHPs are further purified by different column chromatography. Generally, separation of neutral or acidic polysaccharides is usually carried out by ion exchange chromatography. The chromatographic medium mainly includes DEAE-Sephadex A-25, DEAE-Sephadex CL-6B, and DEAE-cellulose 52. For gel filtration chromatography, Sephadex G-100 or Sephadex G-200 are usually used as chromatographic media to separate polysaccharides with different molecular weights. Then, the purified fractions from MFHPs are prepared by concentration, dialysis, and freeze-drying. Finally, the Bradford’s method and phenol-sulfuric method are used to measure the contents of polysaccharides and protein. A schematic diagram of the extraction and purification process of MFHPs is presented in Figure 1.

## 3. Structure–Activity Relationship of MFHPs

Natural MFHPs, as high molecular polymers isolated from MFH materials, have complex and diverse structures. To better understand MFHPs, it is necessary to describe their structural characteristics in detail, including molecular weight (Mw), monosaccharide composition (MC), monosaccharide sequence, type of glycosidic bonds, configuration and spatial conformation. Currently, a series of advanced analytical technologies, namely, Fourier transform infrared spectrometer (FT–IR), nuclear magnetic resonance (NMR), high performance liquid chromatography (HPLC), gas chromatography-mass spectrometer (GC–MS), and other methods, are used to determine the fundamental structural characteristics of MFHPs. Table 2 summarizes the existing information on the basic structural characteristics of MFHPs. The relative Mw size has a substantial influence on the biological activities of MFHPs. Li et al. (2020) prepared polysaccharides from *Ganoderma lucidum* that obtained two polysaccharides frictions (GLP-1 and GLP-2) by different purification techniques. It was found that the molecular weights of GLP-1 and GLP-2 are 1.07 × 10^5^ Da and 1.95 × 10^4^ Da, respectively. Moreover, GLP-1 had a stronger immunomodulatory effect than GLP-2 in increasing hematocrit as well as thymic and splenic indices [33]. Many polysaccharides with outstanding biological activities are often linked by (1→3) glycosidic bonds and have a main chain structure with β-(1,3)-D glycosidic bonds, along with some side chains and a certain degree of branching. Zeng et al. (2017) purified a new water-soluble polysaccharide (PEPW80-1) with a Mw of 4.7 kDa from *Phyllanthus emblica* pulp tissues. The experiments in vitro showed that PEPW80-1 exhibited significant immunomodulatory and antioxidant activities [34]. Chen et al. (2017) prepared various phosphorylated derivatives by different purification methods. It was found that the high Mw and extended chain conformation could improve the anti-tumor activity in vitro and in vivo [35]. Liu et al. (2020) obtained a new polysaccharide (APS-2I) with a Mw of 7.2 × 10^5^ Da from *Angelica sinensis* Diels, and it included six monosaccharides, namely, mannose (4%), rhamnose (5%), galacturonic acid (1%), glucose (10%), galactose (23%), and arabinose (39%). Moreover, G-4, a galactosidase digested fraction of APS-2I, showed strong anti-tumor efficacy [36]. Zhang et al. (2016) extracted twenty-six *Codonopsis pilosula* polysaccharides (CPPs) by water extraction and alcohol precipitation method from *Codonopsis pilosula* collected from 26 different habitats. In addition, they studied the correlation between monosaccharide compositions and their in vitro cytotoxic activities. These results showed that all 26 batches of CPPs possessed cytotoxic activities against HepG2 cells. Additionally, galactose, rhamnose, galacturonic acid, arabinose, and fructose were correlated positively with the cytotoxic activities, whereas glucose, xylose, mannose were correlated negatively with the cytotoxic activities [37]. Wang et al. (2019) obtained Ginger polysaccharide (GP) by enzymatic method and its chemical properties and anti-tumor activity were studied. These results showed that GP included L-rhamnose (3.64%), D-arabinose (5.37%), D-mannose (3.04%), D-glucose (61.03%), and D-galactose (26.91%). In addition, GP showed anti-tumor effects owing to its triple helix structure [38,39]. Significantly, the anti-tumor effect of polysaccharides with a triple helix structure from *Dandelion* was shown to be similar [38,39]. Notably, because of the various raw materials and preparation processes, there are substantial variances in the chemical structure of MFHP products. In addition, the previously reported literature on the chemical structure of MFHPs is summarized in Table 2.

**Table 2 molecules-27-03215-t002:** Sources, purification methods, and structural features of MFHPs.

Sources	Purification Methods	Compound Name	Monosaccharide Composition	Analysis Technique	Chemical Structure	References
Chinese yam	Ultrafiltration	HSY-I, HSY-Ⅱ, HSY-Ⅲ	HSY-I: GluA:Gal = 1.86:5.19 HSY-Ⅱ: GluA:Ara:Rha:Glu:Gal = 0.81:1.24: 2.35:66.79:28.81 HSY-Ⅲ: Man:Glu:Gal = 13.20:12.79:74.0	HPGPC, GC, FT–IR	1	[40]
Ginger	DEAE-52	HGP, EGP1, EGP2, UGP1, UGP2	HGP: Glu EGP1: Man, Ara, Glu, Gal EGP2: Man, Rha, Ara, Gal, Xyl, Glu UGP1: Ara, Gal, Glu, Man UGP2: Man, Rha, Ara, Gal, Glu, Xyl	FT–IR, NMR	ND	[19]
Pueraria lobata roots	DEAE-52	PS-D1, PS-D2, PS-D3	PS-D1: Glu:Fru = 24.4:1.0 PS-D2: Glu:Fru:Ara = 54.5:1.0:1.0 PS-D3: Glu:Fru:Gal:Ara = 61.0:1.0:2.7:2.4	FT–IR, UV	ND	[41]
Raspberry	Macroporous resin, Sephadex G-100	RCP-II	GalA:Glu:Ara:Xyl:Rha:Gal = 1.00: 0.44:1.19:0.52:0.55:1.90	GC, UV, FT–IR, NMR	ND	[24]
Angelica dahurice roots	DEAE-52, Sephadex G-100	ADPs-1a, ADPs-1b, ADPs-2, ADPs-3a, ADPs-3b	ADPs-1a: Glu, Man, Xyl, Gal = 26.1:0.22:0.31:0.11 ADPs-1b: Ara, Glu, Man, Xyl, Gal = 0.10:15.3:0.07:0.26:1.37 ADPs-2: Ara, Glu, Man, Xyl, Gal, Rha = 1.79:15.8:0.40:0.35:5.59:0.34 ADPs-3a: Ara, Glu, Man, Xyl, Gal, Rha = 2.01:1.68:0.41:0.13:4.97:1.06 ADPs-3b: Ara, Glu, Man, Xyl, Gal, Rha = 0.36:13.5:0.09:0.25:1.59:0.18	HPSEC, GC, FT–IR, NMR	2	[42]
Imperial Chrysanthemum	Sephadex G-200	ICP-1	Rha, Ara, Man, Glu, GluA, GalA = 1:0.70:1.14:1.48:0.81:1.67	FT–IR, NMR, SEM, HPGPC	3	[43]
Lotus	Sephadex G-100	LLWP-1, LLWP-3	LLWP-1: Rha, Glu, Gal, Ara, GalA = 7.0:6.0:28.0:24.8:26.4 LLWP-3: Rha, Glu, Gal, Ara, Man, GalA = 6.6:8.9:15.0:9.8:6.1:47.2	HPAEC–PAD, FT–IR,	ND	[44]
Turmeric	DEAE-52	TPS-0, TPS-1, TPS-2, TPS-3	TPS-0: Ara, Gal, Glu TPS-1: Ara, Gal, Glu TPS-2: Xyl, Glu, Gal, Ara, Rha, GalA, GluA TPS-3: Rha, Glu, Gal, Ara, Xyl, GalA, GluA	HPGPC, FT–IR, GC–MS, NMR, SEM	4	[16]
Platycodon grandiflorus	Ultrafiltration	LMw-PGP	ND	HPGPC, FT–IR	ND	[45]
Ganoderma lucidum	QFF anion-exchange column	GLP-1, GLP-2	GLP:Man:Glc:Gal:Fuc = 4.9:63.5:26.2:5.4 GLP-2: Man:Glc:Gal = 1.6:90.6:7.8	Agilent ZORBAX Eclipse XDB-C18 column, HPGPC, FT–IR, NMR	5	[33]
Longan	DEAE-Sepharose	LPIIa	Rha, Rib, Ara, Xyl, Glu, Gal = 1.05:1:22.88:1.01:2.59:34.58	GC–MS, APC, FT–IR, NMR,	6	[46]
Dandelion	DEAE-Sepharose	PD1-1	Glu, Man	HPGPC, GC–MS, UV, FT–IR, NMR	7	[39]
Cistanche tubulosa	Ua-ternary ammonium salt precipitation	CTP	Rha, Man, Glu, Gal = 2.18:1:28.29:1.43	FT–IR, CD, SEM,	ND	[47]
Mulberry leaf	Sephadex G-100	MLP	Ara, Xyl, Glu, Rha, Man = 1:2.13:6.53:1.04:8.73	HPSEC, HPLC, FT–IR	ND	[48]
Sea buckthorn	Sephacryl S-200 column	SP0.1-1	Ara, Glu, Gal, Man = 11.2:2.3:1.9:1	HPGPC, GC–MS, NMR,	8	[49]
Amomum villosum Lour	DEAE-52, Sephadex G-100	AVPG-1, AVPG-2	AVPG-1: Glucose, Galactose, Xylose, Arabinose, GluA = 73.11:10.29:6.21:8.83:1.57 AVPG-2:Rha, Glu, Gal, Xyl, Ara, Glu GalA = 3.11:40.23:17.82:12.53:15.81:3.99	HPSEC, SEM, GC–MS, NMR, FT–IR	9	[50]
Gastrodia elata	Membrane separation	GEP-1	Glc	GC–MS, FT–IR, NMR, SEM	10	[51]
Dendrobium officinale	DEAE-52, Sephacryl S-300	DOPA-1, DOPA-2	DOPA-1: Man:Glu = 5.8:1 DOPA-2: Man, Glu = 4.5:1	HPGPC, FT–IR, GC–MS, NMR	11	[52]
Phyllanthus emblica	DEAE-52, Sephadex G-100	PEPW80-1	Rha, Arab, Gal = 3.02:1.00:4.23	HPAEC–PAD, GC, FT–IR, NMR, SEM	12	[34]
Houttuynia cordata	Sephacryl S-300, DEAE Cellulose	HCA4S1	Rha, GalA, Gal, Ara = 15.6:17.5:41.2:25.7	HPGPC, UV, GC, FT–IR, NMR,	13	[53]
Polygonatum sibiricum	DEAE-Sepharose	PSP1, PSP2, PSP3, PSP4	PSP1: Man:Glu:Gal = 14.96:2.13:82.91 PSP2: Rha:Glu:Gal:Xyl = 20.54:2.06:74.37:3.03 PSP3: Man:Rha:Glu:Gal:Xyl = 1.38:57.69:2.02:37.17:1.74 PSP4: Man:Rha:Gal:Xyl = 2.00:72.63:20.74:4.63	HPSEC, FT–IR, NMR	ND	[54]

Note: 1. The ratios of 1→2, 1→4 in HSY-I, HSY-II, HSY-III glycosidic bonds were 5.79: 15.4: 9.39: 4.3. 2. α-type glycosidic linkage. 3. ICP-1 were composed of (1→), (1→4) and (1→6) glucose, (1→5) arabinose, (1→4) galacturonic acid and (1→3,6) mannose. 4.TPs-0 comprised a main chain ofα-Ara*f*- (1→4) -α-Glc*p*- (1→3) -α-Ara*p*- (1→3) -β-Gal*p*- (1→3,6) -α-Gal*p*- (1→5) -α-Ara*f*- (1→3) -β-Gal*p*- (1→R. 5. GLP-1: →6)-β-D-Glcp-(1→, →6)-α-D-Galp-(1→, and→3)-β-D-Glcp-(1→residues. GLP-2: β-D-glucan that possessing→6)-β-D-Glcp-(1→and→3)-β-D-Glcp-(1→residues packaged into a spherical conformation. 6. (1→3,4)-linked-α-Rha *p*, (1→4)-linked-β-Gal*p*, (1→6)-linked-β-Gal*p*, and (1→3,6)-linked-β-Gal*p.* 7. α-d-Man/Glc*p*-(1→and→1)-β-d-Man/Glc*f*-(2→glycosidic linkage conformations. 8. 1, 4-linked-α-D-Glcp, 1, 4, 6-linked-α-D-Glcp and 1, 4-linked-α-D-Manp residues as the back bone. And the side-chains comprised of 1, 3, 5-linked-α-L-Araf, 1, 5-linked-α -L-Araf, terminalα-Araf and 1, 4-linked-β-D-Galp. 9. AVPG-1: →4)-α-D-Glc*p*-(1→3,4)-β-D-Glc*p*-(1→4)-α-D-Glc*p*-(1→AVPG-2: →4)-α-D-Glc*p*-(1→3,6)-β-D-Gal*p*-(1→4)-α-D-Glc*p*-(1→. 10. 1,3,6-linked-α-Glcp,1,4-linked-α-Glcp, 1,4-linked-β-Glcp and 1,4,6-linked-α-Glcp. 11. 1,4-linked-β-D-Man*p* and 1,4-linked-β-D-Glc*p* with O-acetyl groups. 12. Backbone is composed of (1,3)-linked-β-L-Rhamnose and (1,3,6)-linkage-β-D-galactose, with two branch chains of (1,4)-linked-α-D-galactose and (1,6)-linked-β-D-galactose and terminated with 1-α-L-arabinose. 13. 1,4-linked α-D-GalA and 1,2,4-linkedα-L-Rha. The latter was substituted at C-4 position by 1,4 linked, 1,6-linked β-Galp, or Teminal linked β-Gal.

## 4. Biological Activities of MFHPs

As a popular active component of MFH materials, MFHPs have significant pharmacological activities and health effects. Recently, with the in-depth study of MFHPs, researchers have obtained a comprehensive understanding of their biological activities. The biological activities of MFHPs mainly include hypoglycemic, hypolipidemic, antioxidant, anti-tumor, immunomodulating, protective effect of the intestinal barrier, anti-inflammatory, antiviral, and other activities (Figure 2). MFHPs’ biological actions are inextricably linked to their structural features. Hence, clarifying the relationship between the structure and efficacy of MFHPs is of great significance for improving the performance of MFHPs and developing future drugs. However, the current reports on the relationship between MFHP structural and biological activities are limited, and no consensus has been established due to the complex structure of MFHPs. Therefore, we can only review this on the basis of existing research reports.

### 4.1. Hypoglycemic Activity

Diabetes mellitus is a group of metabolic disorders characterized by hyperglycemia. Hyperglycemia can cause long-term damage and dysfunction in various organs, including the kidneys, eyes, blood vessels, heart, and nerves. Diabetes has now become a significant socioeconomic burden in many nations as a prevalent metabolic illness. The direct cause of diabetes is the dysfunction of islet cells or the insensitivity of the body to insulin, which leads to the decrease of insulin secretion, and the low efficiency of glucose utilization and storage in the blood [55]. Wang et al. (2019) used several isolation methods to isolate a new polysaccharide (MFP4P) with hypoglycemic action from mulberry fruit pulp. These results show that MFP4P with a Mw of 198.2 kDa contained galactose, arabinose, glucose, galacturonic acid, and rhamnose. In addition, MFP4P and zinc complexes (MFP4P-Zn) showed higher hypoglycemic activity than MFP4P at the same concentration, indicating that MFP4P has potent insulin-sensitizing effect in vitro, boosting insulin secretion, and encouraging pancreatic cell proliferation [56]. In diabetic rats, mulberry polysaccharide fractions (MFP50 and MFP4P) have been demonstrated to control the insulin signaling system via activating the PI3K/AKT pathway [56,57,58]. Jiang et al. (2018) isolated a novel polysaccharide (YZ-2) with a Mw of 5345.3 Da from *Polygonatum odoratum* (*Mill.*) *Druce* and studied their hypoglycemic activity. In T2DM mice, it was revealed that YZ-2 may lower fasting blood glucose levels, serum TG, TC, and LDL-c concentrations, as well as fasting insulin levels, while improving glucose tolerance ability [59]. Zhu et al. (2016) isolated *Astragalus* polysaccharides (APs) with a high Mw of 693 kDa, and APs demonstrated a dose-dependent inhibitory impact on α-glucosidase [60]. Notably, similar results were observed on the inhibiting α-amylase and α-glucosidase activities of *Pueraria lobata* polysaccharides [61], mulberry fruit polysaccharides [62,63,64,65], and *Pacific Oyster* polysaccharides [66].

### 4.2. Hypolipidemic Activity

Hyperlipidemia can lead to a range of diseases such as atherosclerosis, stroke, coronary heart disease, myocardial infarction, diabetes, and kidney failure [67]. At present, the main lipid-lowering drugs used in clinical practice are statins and Bette drugs, whereas these drugs may cause liver injury, rhabdomyolysis, and diabetes [68,69]. Hence, it is necessary to develop new lipid-lowering drugs for clinical application. Xu et al. (2019) prepared natural polysaccharides (GLP) and degraded polysaccharides (GLPUD) from *Ganoderma lucidum*, and measured their hypolipidemic activity. GLPUD was discovered to have greater hypolipidemic action than GLP. GLPUD was more effective than the GLP for decreasing AI, TG, TC, and LDL-C, increasing HDL-C, reducing MDA content, and improving the GSH-Px in mice serum, enhancing SOD activity in liver [70]. Pan et al. (2017) prepared an important polysaccharide from *Astragalus membranaceus* (AMP) and investigated its hypolipidemic activity. The results showed that AMP (50, 100, and 150 mg/kg) could reduce the levels of TG, TC, and LDL-C in plasma. In addition, AMP could enhance the levels of fecal fat, TG, and HDL-C of rats compared with the control group, indicating that AMP has strong hypolipidemic activity and can be used in the treatment of hyperlipidemia [6]. From a lotus leaf, Zeng et al. (2017) extracted lotus leaf selenium (Se)-polysaccharide (LLP), and evaluated its antioxidant activity and insulin resistance in pregnant rats. The results showed that LLP (50 and 100 mg/kg) could restore the weight loss of pregnant rats, placentas, and fetal rats in gestational diabetes mellitus (GDM) rats, and LLP could decrease the levels of FBG and FINS in GDM rats, whereas it could increase hepatic glycogen content. In addition, LLP could reduce TC, TG, and LDL-C, cholesterol levels except for HDL-C cholesterol level, and increase the activities of SOD, CAT, GSH-Px, and GSH in liver tissues, suggesting that LLP may be a promising candidate drug or health food for GDM treatment [71]. These natural MFHPs have high lipid-lowering biological activities and low toxic and side effects. They are expected to become new lipid-lowering drugs with high efficiency and low toxicity.

### 4.3. Antioxidant Activity

Oxidative stress and imbalance of free radical metabolism are integral parts of the pathological process of most diseases. The enzymatic antioxidant system and the non-enzymatic antioxidant system jointly maintain the redox balance in the body. The enzyme antioxidant system is mainly composed of endogenous antioxidant enzymes (CAT, GSH-Px, and SOD) [72]. The antioxidant capabilities of MFHPs are usually assessed by various free radical tests, such as DPPH, OH, O^2−^, and ABTS free-radical scavenging in vitro. Zou et al. (2020) prepared two pectic polysaccharides (CPSP-1 with a Mw of 13.1 kDa and CTSP-1 with a Mw of 23.0 kDa) and evaluated their antioxidant activity. The results showed that CTSP-1 and CPSP-1 have antioxidant action via the intestinal cellular antioxidant defense mechanism, which might protect cultured intestinal cells from oxidative damage [73]. Nuclear factor-E2-related factor 2 (Nrf2) plays a key role in oxidative stress. Nrf2 interacts with Keap1 during non-oxidative stress. The binding between the two is rapidly destroyed and remains low in the cytoplasm when oxidative stress occurs. During oxidative stress, Nrf2 enters the nucleus, interacts with antioxidant response elements (ARE), and regulates the activities of CAT, GSH-Px, and SOD [74]. Qin et al. (2018) prepared the best selenizing *Codonopsis pilosula* polysaccharides (SCPPS_5_) and investigated their protective effect on H_2_O_2_-induced oxidative damage of RAW264.7 cells. It was found that SCPPS_5_ could significantly decrease the production of ROS, whereas SCPPS_5_ could obviously increase the expression levels of Nrf2, NQO1, Keap1, SOD, and HO-1 in H_2_O_2_-induced RAW264.7 cells, indicating that SCPPS_5_ can attenuate cellular oxidative stress levels via inhibiting the Keap1-Nrf2/ARE signaling pathway [75]. Liu et al. (2016) isolated and purified three water-soluble polysaccharides, namely, PS-D1, PS-D2, and PS-D3, and evaluated their antioxidant activity. It was found that PS-D1 with a Mw of 62 kDa included glucose (24.4%) and fructose (1.0%), whereas PS-D2 with a Mw of 159 kDa and PS-D3 with a Mw of 385 kDa were both acidic heteropolysaccharides comprising glucose, galactose, fructose, and arabinose. Furthermore, PS-D3 at 3.0 mg/mL had the strongest scavenging capacities against OH (79.6%) and DPPH (60%) radicals, as well as the highest antioxidant activity with a Trolox equivalent antioxidant capacity of 53.8 μmol Trolox/g, indicating that PS-D3 with prominent antioxidant capability had glycation inhibition activity in vitro [41]. Gong et al. (2017) extracted the LBP by HWE and then purified LBP via ion-exchange column chromatography to obtain three fractions (LBP-I-1, LBP-I-2, and LBP-I-3) and evaluated their antioxidant activity. Results showed that LBP-I-3 could most obviously improve macrophages NO, acid phosphatase, and phagocytic capacity [76]. Chen et al. (2018) determined the preventive effects of LBP against lipopolysaccharide (LPS)-induced ARDS and potential molecular mechanisms. The results showed that LBP could reduce LPS-induced lung inflammation and pulmonary oedema in vivo. In vitro, LBP could inhibit caspase-3 activation and ROS production, reversing LPS-induced apoptosis, oxidative stress and decreases in cell viability. Furthermore, LBP could dramatically reduce LPS-induced NF-κB activation and reversed cytochrome c release [77,78]. Liu et al. (2015) also investigated the effect of LBP in protecting ARPE-19 cells from oxidative stress-induced apoptosis. It was found that LBP could decrease ROS levels via inhibiting caspase-3 activation and improving antioxidant enzyme activity [77,78]. Zhuang et al. (2018) found that polysaccharides from *Angelica sinensis* could decrease the levels of iNOS, ROS, and MDA, and increase the activities of SOD and CAT, suggesting that polysaccharides can be used in the treatment of osteoarthritis [79]. Wang found that CPPs could reduce the levels of IL-6 and CORT in a rat model with kidney-yin deficiency stimulated by hydrocortisone [80]. Moreover, raspberry polysaccharides [24], *Platycodon grandiflorus* polysaccharides [45], *Ganoderma lucidum* polysaccharides [70], mulberry polysaccharides [65], *Cyperus rotundus* polysaccharides [81], and *Dendrobium heliotrope* polysaccharides [52] showed good antioxidant activity. Figure 3 illustrates the antioxidant mechanism of MFHPs.

### 4.4. Anti-Tumor Activity

The World Health Organization reported that cancer causes 8.2 million deaths every year, accounting for 13% of the total deaths in the world [82]. Despite significant advancements in the identification and treatment of human malignant tumors, the long-term prognosis is still poor. Therefore, viable anti-tumor medications that are both safe and effective must be created. As of now, MFHPs have been confirmed to show great anti-cancer potentials, and they are considered as promising candidate drugs for cancer treatment [83]. According to research, MFHPs and their related derivatives can be used for cancer treatment alone or in conjunction with other bioactive components.

Bai et al. (2018) isolated two water-soluble polysaccharides from Codonopsis pilosula Nannf. var. modesta (Nannf.) L.T. Shen (CPP1a with a Mw of 1.01 × 10^5^ Da and CPP1c with a Mw of 1.03 × 10^5^ Da) via different purification methods and evaluated their anti-tumor activity. These results show that the sensitivity of HepG2 cells to CPP1a and CPP1c was higher than that of MKN45 cells and HeLa cells. Both CPP1a and CPP1c could inhibit migration, affect cell morphology, and induce apoptosis by affecting the G2/M phase in HepG2 cells, suggesting that CPP1a and CPP1c could increase apoptosis rate through up-regulation of Bax/Bcl-2 and caspase-3 [84]. Mao et al. (2011) investigated the effect of LBP on SW480 and Caco-2 cells and its possible mechanisms. It was discovered that LBP could inhibit the growth of Caco-2 and SW480 cells in a dose-dependent way and arrest at the G0/G1 phase, suggesting that LBP can be used as a candidate natural anticancer drug [85]. Miao et al. (2010) extracted LBP and evaluated its potential anticancer activity. These results indicate that LBP could reduce survival rate of SGC-7901 and MGC-803 cells, and arrest at the G0/G1 and S phases, respectively, indicating that LBP could involve in the induction of anticancer cell cycle [86]. Zhang et al. (2005) prepared LBP and evaluated its anticancer activity. It was found that LBP could inhibit QGY7703 cell growth, enhance apoptosis induction, and arrest at the S phase as well as increase Ca^2+^ concentration, implying that the increase of intracellular calcium in QGY7703 cell cycle arrest and apoptosis system may be involved in the antiproliferative activity of LBP [87]. Ma et al. (2020) extracted a homogeneous polysaccharide from Hawthorn (HPS) and determined its anticancer effects. The result suggests that HPS could inhibit HCT116 cells from proliferating, arrest the cell cycle, and increase the apoptosis rate of HCT116 cells. Moreover, HPS could down-regulate the expression levels of Cyclin A1/D1/E1 and CDK-1/2, and up-regulate the expression levels of Fas and caspase-3/7/8/9, TRADD, TNF-R1, and FADD, indicating that HPS could become a potential candidate for functional food for cancer patients in the future [88]. Han et al. (2018) identified and purified a water-soluble pectic polysaccharide HCA4S1 with a Mw of 21.7 kDa, and evaluated their anti-lung cancer activity. These results show that HCA4S1 could inhibit the proliferation of A549 cells by inducing cell cycle arrest and apoptosis [53]. Polysaccharides from *Astragalus membranaceus* significantly inhibited the growth of H22 cells in vivo by increasing serum cytokine (IL-2, TNF-α, and IFN-γ) levels, improving immune cell (lymphocytes, NK cells, and macrophages) activities, and inducing cell apoptosis [89]. In addition, MFHPs could inhibit tumor proliferation by inducing apoptosis [90], promoting NO release [91], and affecting the gut microbiota lineage [92]. Especially, more research is needed to elucidate the specific mechanism of anti-tumor activity of MFHPs.

### 4.5. Immunomodulating Activity

Numerous investigations have revealed that MFHPs have significant immunomodulatory activity. Zhang et al. (2017) extracted a pectic polysaccharide from *Codonopsis pilosula* (CPP1c) and evaluated its immune-modulating activities in vitro and in vivo. These results show that CPP1c could improve lymphocyte proliferation, modulate the percentage of CD^4+^, CD^8+^, CD^28+^, and CD^152+^ T cells and promote the levels of IL-2, TNF-α, and IFN-γ, indicating that CPP1c may promote T cell activation and play an immunostimulatory role through TCR/CD28 signaling pathway [93]. Fu et al. (2017) investigated microflora regulation and immunomodulating activity of CPP and found that CPP (50, 100, 200 mg/kg/d) could restore the levels of IL-2, IL-10, IFN-γ, and serum IgG. Moreover, CPP was important in preventing mucosal immune damage and inhibiting pathogen colonization, implying that CPP could be considered as a potential source of natural immune modulators [94]. Wang et al. (2016) found that *Angelica sinensis* polysaccharides (CAPA70) could promote lymphocyte proliferation and increase IL-2, IL-6, IFN-γ, and TNF-α, as well as the percentage of CD^3+^ and CD^56+^ cells in peripheral blood lymphocytes [95]. Hou et al. (2016) extracted and purified lily polysaccharide (LP) and evaluated its immunomodulating activity. The results show that sLP6 could enhance lymphocyte proliferation and promote the levels of IL-2, IL-6, and IFN-γ, suggesting that polysaccharide selenizing could obviously increase the immune-enhancing activity of LP [96]. *Caulophyllum* polysaccharides could inhibit NO production, IL-6, and TNF-α levels, and enhance its phagocytic capacity [50]. Furthermore, the immunomodulating activity of *Gastrodia elata* polysaccharides was shown to be similar [97] to *dendrobium* polysaccharides [52], and *American ginseng* polysaccharides [98]. In summary, the above results suggest that water-soluble MFHPs may be candidate drugs for the treatment of related immune diseases.

### 4.6. Protective Effect of the Intestinal Barrier

The gut is a natural barrier that maintains the balance of the internal environment and impedes pathogens and toxins. The intestinal barrier is mainly composed of four parts (biological, chemical, mechanical, and immune barriers). Among them, biological, mechanical, and immune barriers are the main components of intestinal mucosal immunity [99,100]. MFHPs maintain the stability of the intestinal environment by protecting intestinal biological, mechanical, and immune barriers (Figure 4). *Pueraria lobata* polysaccharides could increase probiotic abundance, reduce isovaleric acid concentration, and attenuate colonic lesions caused by antibiotic-associated diarrhea [101]. Mulberry polysaccharides have the potential to enhance the number of bacteroides and reduce the number of firmicutes as shown in in vitro fermentation experiments. In addition, some intestine beneficial bacteria (such as lactobacillus and bifidobacterium) may absorb mulberry polysaccharides and create short chain fatty acids, which has a good impact on intestinal ecology [102]. Bai et al. (2020) isolated and purified polysaccharides (LPⅡa with a Mw of 159.3 kDa) from longan pulp, and evaluated their intestinal barrier protection and anti-inflammatory activity. The results show that LPⅡa could inhibit the production of TNF-α, NO, and PGE2, and suppress iNOS and COX-2 mRNA expression. In addition, LPⅡa could reduce the expression of intestinal tight junction channel protein Claudin-2 and increase the expression of tight junction ZO-1 protein in Caco-2 cells [46]. Honey polysaccharides could reverse the reduction of villus length/recess depth ratio and the number of goblet cells caused by cyclophosphamide, and increase the secretion of ZO-1 and Mucin-2 to protect the intestinal mechanical barrier [103]. Wang et al. (2020) found that APS could regulate the abnormal expression of MyD88 and NF-κB and inhibit the production of IL-1β and TNF-α, thus protecting the intestinal epithelial barrier from damage [104]. Liang et al. (2017) found that *Dendrobium* polysaccharides could reduce the loss of mitochondria in tumor-infiltrating CD^8+^ cytotoxic T lymphocytes and PD-1 expression, improve T cell metabolic function, regulate the tumor micro-environment, protect the intestinal barrier, and inhibit the development of colorectal cancer [103]. *Ganoderma lucidum* polysaccharides could modulate intestinal immune barrier function by increasing NF-κB and sIgA expression and serum cytokine IL-2, IL-4, and IFN-γ levels [105].

### 4.7. Anti-Inflammatory Activity

Inflammation is a vital part of the human body’s defense system, and it plays a positive role in preventing the spread of pathogenic microorganisms, diluting toxins, destroying inflammatory factors, removing necrotic tissue and repairing damaged tissue. However, excessive inflammatory response can lead to organ tissue and cell degeneration and necrosis in some cases. Exudation and edema of intracranial inflammation can lead to brain hernia. In addition, long-term chronic inflammatory stimulation can also induce tumors. Therefore, controlling inflammatory responses is the key to the treatment of inflammatory diseases. CPPs could enhance phagocytosis of alveolar macrophages in mice with chronic obstructive pulmonary disease, lower TNF-α, IL-6, and IL-8 levels in serum and bronchoalveolar lavage fluid, inhibit inflammatory responses and improve airflow limitations [106,107]. *Flavopiridium* polysaccharides had a strong protective effect against acute kidney injury in transgenic rats and could reduce the expression of NGAL or KIM-1 genes in kidney tissues, and inhibit the production of the p38 MAPK/ATF2 signaling pathway as well as the inflammatory mediators (IL-1β, IL-6, and TNF-α) [108]. *Ganoderma lucidum sulfated* polysaccharides (GLPss58) could inhibit the levels of IFN-γ and TNF-α. Moreover, GLPss58 could restrain inflammatory responses [109]. LPIIa could inhibit the production of TNF-α, NO, IL-6, and PGE2 in RAW264.7 macrophages and suppress COX-2 and iNOS gene expression [46]. In conclusion, MFHPs have good anti-inflammatory activity, but their mechanisms still need to be further investigated.

### 4.8. Antiviral Activity

The antiviral mechanism of natural drug polysaccharides mainly includes direct inhibition and killing of virus, inhibition of virus biosynthesis and proliferation, blocking virus adsorption and entry into cells, and immune regulation of host. Liu et al. (2015) prepared SP_9_-sCP_1_ with polysaccharide (PS) and sulphated polysaccharide (sPS), which showed good viral killing and inhibition of viral antigen expression. This suggests that SP_9_-sCP_1_ has high antiviral efficacy and could be used as a new antiviral drug [110]. Xing et al. (2021) investigated the antiviral activity of *Platycodon grandiflorus* polysaccharides (PGPS) and virus-induced autophagy in the anti-PRV effect of PGPSt in PK-15 cells. In addition, this paper evaluated the effect of PGPSt on PRV replication and virus-induced autophagy. These results showed that PGPSt could reduce PRV replication and activate the Akt/mTOR signaling pathway that is inhibited by PRV infection, implying that PGPSt can prevent PRV infection and inhibit PRV replication by attenuating PRV-induced autophagy [111]. Lee et al. (2015) investigated the anti-porcine epidemic diarrhea virus (PEDV) impact and mode of action of polysaccharide from *Ginkgo biloba* exocarp in vitro. When compared to a positive control, these results show that the polysaccharides had significant antiviral activity against PEDV, limiting the creation of a noticeable cytopathic impact. In addition, polysaccharides inactivated PEDV infection in dose- and temperature-dependent manners. Hence, polysaccharides with effective inhibition of virus attachment and entry into the life cycle of PEDV are good candidates for the development of antiviral drugs [112]. All in all, MFHPs can directly kill viruses, inhibit virus adsorption, invasion and replication, and activate the immune system. Polysaccharides, as the main active components of MFH, may have more antiviral mechanisms. The research on the antiviral mechanisms of MFHPs still needs to be continued.

### 4.9. Other Activities

Except for the above biological activities, MFHPs showed other biological activities. *Astragalus* polysaccharides could reduce apoptosis of cardiomyocytes due to ischemia-reperfusion injury and reduce cardiomyocyte edema [113]. To simulate renal ischemia-reperfusion injury, 10 mg/kg *Codonopsis pilosula* polysaccharides was given by gavage 10 d before renal ischemia. It was found that the activities of creatinine, serum urea nitrogen, TNF-α, aspartate aminotransferase, and lactate dehydrogenase decreased significantly. *Codonopsis pilosula* polysaccharide was found to reduce renal injury by preventing the release of proinflammatory factors and TNF-α [114]. Mulberry polysaccharides have been shown to boost alcohol dehydrogenase activity, which could protect against acute and subacute alcoholic liver injury [115,116,117]. Furthermore, polysaccharide components (MFP-II) could reduce oxidative stress and palmitic acid-induced liver fat toxicity via activating the Nrf2/ARE signaling pathway [118]. The histopathological study of mouse liver showed that jujube polysaccharides could prevent liver injury induced by carbon tetrachloride and ethylamine. Furthermore, studies have proved that jujube polysaccharides could reduce the content of MDA by increasing the activities of GSH-Px and SOD to prevent liver injury [119]. Zhang et al. (2018) isolated and purified polysaccharides from *Poria cocos (Schw.) Wolf* to obtain two fractions (PCWPW with a Mw of 37.154 kDa and PCWPS with a Mw of 186.209 kDa) and evaluated their antidepressant activities. Two fractions (PCWPW and PCWPS) were discovered to have antidepressant properties and could inhibit ConA-stimulated T cell proliferation in a dose-dependent manner. Moreover, PCWPS could protect PC12 cells from H_2_O_2_-induced damage and inhibit B cell proliferation induced by LPS, suggesting that PCWPW and PCWPS may be candidates for the development of antidepressants or immunosuppressants in the food and pharmaceutical industries [120]. Wang et al. (2017) prepared five polysaccharides from *Angelicae dahuricae Radix* and measured their procoagulant activity. The results show that ADPS-1b and ADPS-2 could exert procoagulant activity by endogenous and exogenous pathways. The procoagulant activity of ADPS-3a was linked to the endogenous pathway and increased fibrinogen content [42]. Wang et al. (2020) modified and obtained three fractions from Ginger polysaccharides, namely SGP, SGP1, and SGP2, and evaluated the internal and external pathways of their coagulation effect. In the determination of activated partial thromboplastin time (APTT), the coagulation time of SGP2 or SGP at the concentration of 2 mg/mL was 41.42 min or 38.01 min, respectively, which was about 1.33 and 1.22 times that of normal saline. After adding 2 mg/mL GP, the prothrombin time (PT) increased by 1.22 times compared to the normal saline control group. However, no coagulation inhibition was detected in the thrombin time test even at the significantly prolonged concentrations of APTT and PT. Therefore, Ginger polysaccharide may be used as anticoagulant and therapeutic reagent for thrombosis [121]. Radiotherapy is the most commonly used clinical treatment for malignant tumors. However, unnecessary ionizing radiation may cause harmful effects on human health. Nowadays, mounting pieces of evidence have confirmed that natural polysaccharides and their derivatives have radiation protection effects [122]. Moreover, current studies have illustrated that MFHPs showed potential radioprotective activity. Chen et al. (2019) prepared SNAAP composed of Glu (1%) and Man (7%), and evaluated its effects on glucose metabolism and pancreas of 60Co-γ-radiated Kunming mice. These results showed that SNAAP administration could decrease radiation-induced glucose metabolism disturbance [123]. Overall, these findings suggest that MFHPs could be used as natural anti-radiation drug candidates. Notably, more clinical trials are demanded to verify the reliability of MFHPs as radiation protective agents.

All in all, previous research has confirmed that MFHPs have various biological activities, consistent with other natural polysaccharides. The detailed structure–effect relationship will contribute to the development of potential health products and clinical drugs from MFHPs. Therefore, many studies are urgently needed to clarify the structure–activity correlation of MFHPs.

## 5. Concluding Remarks and Prospects

Many researchers have contributed valuable discoveries to the scientific research of MFHPs, indicating that MFHPs have considerable medical and nutritional value. Furthermore, MFHPs may serve as a potential treasure trove for pharmaceuticals, functional foods, and cosmetic additives in the future. This paper mainly reviewed the research progress in the extraction, separation, purification, structural characteristics, and biological activities of MFHPs. Despite the fact that MFHP research has progressed significantly, we still need to exert continuous efforts to solve several key technical problems. Firstly, the current extraction and purification methods of MFHPs are inefficient, time-consuming, and complex. Feasible or effective preparation methods for industrialized large-scale extraction and purification of MFHPs are lacking. Secondly, great differences exist in the chemical structure and biological activities of MFHP-based products due to the different raw materials and preparation methods. Hence, the standardized preparation process of MFHPs must be established to ensure the consistency and repeatability of products, which are important for product quality control. Thirdly, the precise chemical structures of MFHPs still lack characterization. In addition, a limited number of studies report the structure–activity relationship of MFHPs and the specific molecular mechanism of their bioactivities. More in vivo experiments and clinical research are needed to verify the reliability of MFHPs usage. Moreover, further studies should focus on assessing the effects of MFHPs combined with other natural bioactive components to promote their clinical application.

To summarize, previous research has established a solid foundation for the potential application of MFHPs in functional foods, health products, and drug therapy. Nevertheless, continuous efforts are still needed to establish preparation methods with development potential, ensure the product quality based on MFHPs, provide precise structural information, and further clarify the structure–activity relationships and potential molecular mechanisms. This review may contribute to a better understanding of MFHPs and provides an important reference for the development and application of MFHPs.

## Figures and Tables

**Figure 1 molecules-27-03215-f001:**
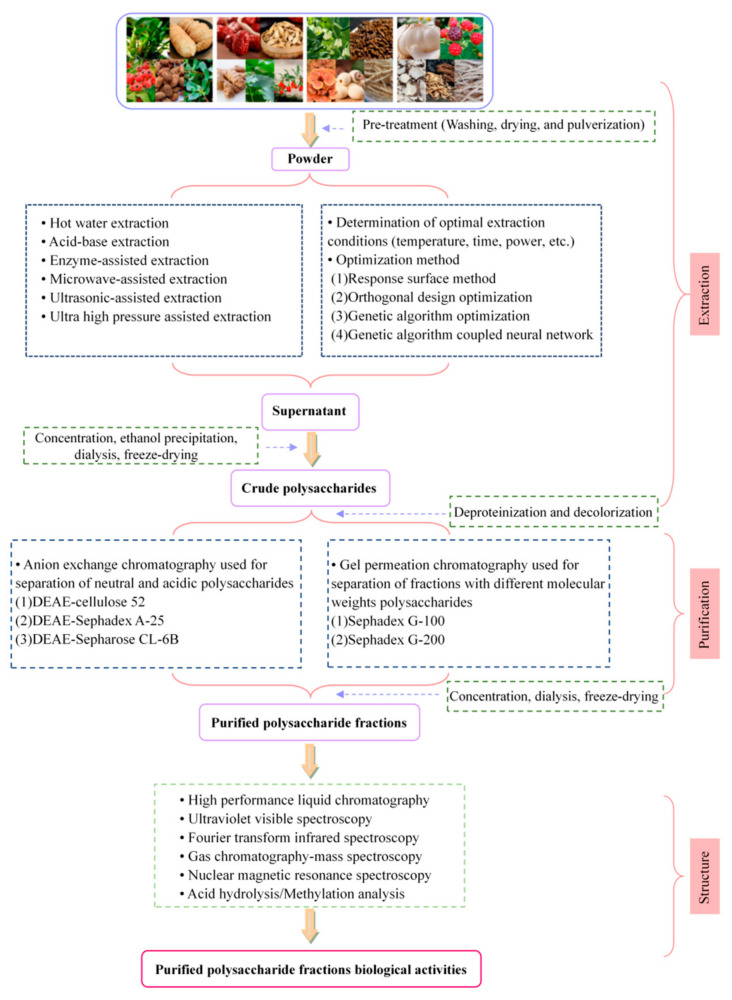
The extraction and purification process of MFHPs.

**Figure 2 molecules-27-03215-f002:**
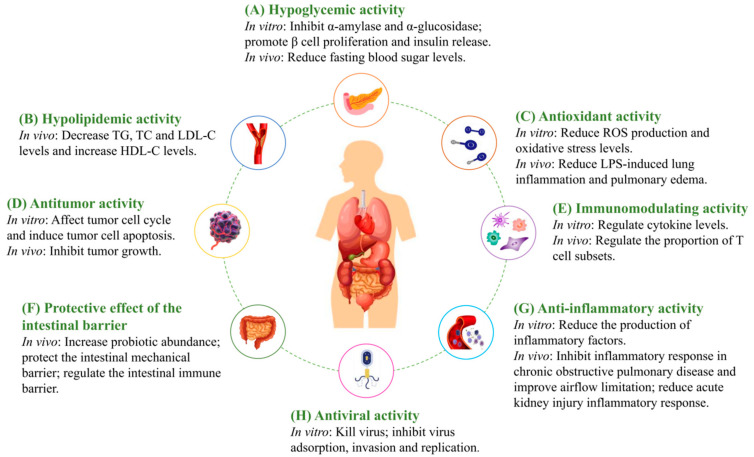
The biological activities and molecular mechanisms of MFHPs.

**Figure 3 molecules-27-03215-f003:**
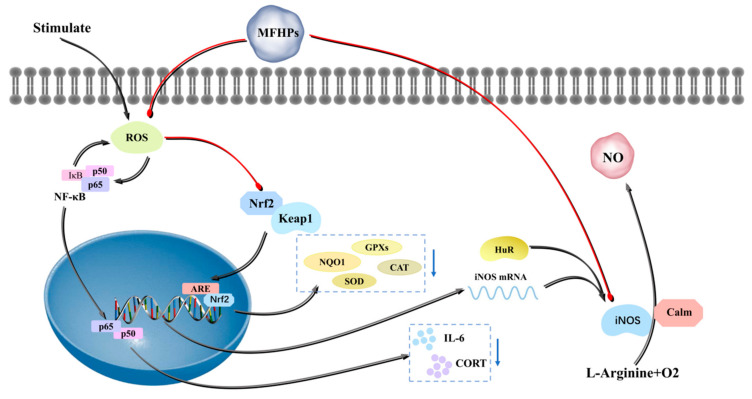
Antioxidant mechanism of MFHPs.

**Figure 4 molecules-27-03215-f004:**
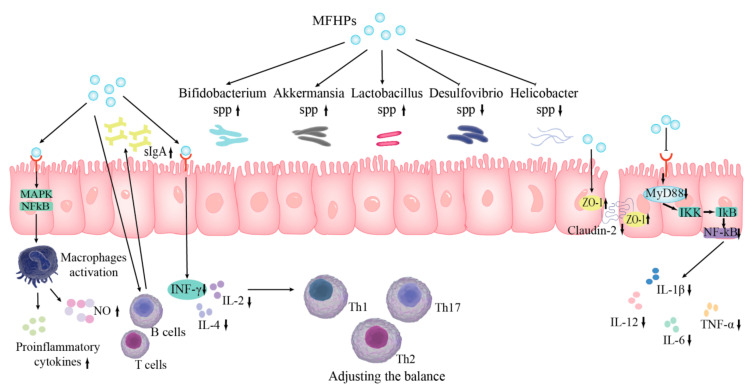
Intestinal barrier protection by MFHPs.

**Table 1 molecules-27-03215-t001:** Extraction methods of MFHPs.

Source	Extraction Methods	Extraction Conditions	Yield /%	Reference
Turmeric	HWE	Extraction temperature of 60–100 °C, liquid-to-material ratio of 5–25 mL/g, extraction time of 1–3 h, and extraction times of 1–3	2.23	[16]
Angelica sinensis	HWE	Liquid-to-material ratio of 5 mL/g, extraction time of 130 min, and extraction times of 5	5.60	[17]
Chinese yam	HWE	Extraction temperature of 100 °C, liquid-to-material ratio of 5 mL/g, extraction time of 3 h	5.71	[18]
Ginger	HWE	Extraction temperature of 100 °C, extraction time of 4 h, and liquid-to-material ratio of 20 mL/g	11.74	[19]
Cassia	HWE	Extraction temperature of 80 °C, extraction time of 3.5 h	5.46	[20]
Lentils	ACE	pH 4, extraction temperature 100 °C, extraction time 90 min	23.30	[32]
Ginger	EAE	20,000 U/g pectinase, 63,000 U/g cellulase, and 62.5 U/g papain, extraction temperature 40 °C, pH 7.0, and extraction time 2 h	7.00	[19]
Raspberry	EAE	Pectinase:cellulase:papain of 2.5:1.7:2.1 (g/g/g), pH 4.0, liquid-to-solid ratio of 10:1 mL/g, extraction temperature of 55 °C, and extraction time of 2.6 h	4.09	[24]
Angelica sinensis	MAE	Microwave power of 500 W, liquid-to-solid ratio of 51 mL/g, and extraction time of 20 min	7.82	[27]
Cassia seed	MAE	Microwave power of 415 W, liquid-to-solid ratio of 51 mL/g, and extraction time of 7 min	8.02	[26]
Coix seeds	UAE	Ultrasonic power 480 W, extraction temperature 80 °C, liquid-to-solid ratio 21 mL/g, and extraction time 16 min	8.34	[29]
Dendrobium officinale	UAE	Ultrasonic power 144 W, extraction temperature 80 °C, liquid-to-material ratio 60 mL/g, and extraction time 3 h	10.29	[30]
Lycium barbarum	UAE	Ultrasonic power 185 W, extraction temperature 73 °C, liquid-to-material ratio 38 mL/g, and extraction time 80 min	12.54	[31]

## Data Availability

Review data are not shared.

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
