# Peer review of "Polysaccharides from Medicine and Food Homology Materials: A Review on Their Extraction, Purification, Structure, and Biological Activities"

_molecules, 2022, doi:10.3390/molecules27103215_

Round 1

Reviewer 1 Report

The review article "Polysaccharides from Chinese Medicinal Materials with Medicine and Food Homology: A Review of Their Extraction, Purification, Structure and Biological Activities" is very interesting and well written. The data was also presented in a good format and referenced accordingly.
In my opinion, the name "CMFHPs" is slightly misleading as research into polysaccharides from various sources has been conducted without regard to traditional Chinese medicine. Also, there is no evidence that these polysaccharides retain the same structure after the preparation of the traditional Chinese medicine using the ingredients mentioned in the review. In addition, the biological activity of various polysaccharides has been directly performed (mentioned in this review) using the extracted polysaccharides but not using traditional Chinese medicine. Most of the data presented in this review article is from plant-based polysaccharides, so it might be a good idea to change the word to medicinal plant-based polysaccharides or something similar instead of “CMFHPs”. I would also like to let you know that I am not against traditional Chinese Medicine, but it is just my suggestion. In addition, there are very few suggestions for improving the article, so I recommend publishing the article after a major revision.

Figure 1: The text in the figure 1 is not well readable, I think it is due to presented in the image format, please make the text well readable.

Figure1: In figure 1, it was written "Podwer" , I think it should be "powder", please correct it. 

Line 105: The font looks different for "Cassia seed" and is not a scientific name, please correct it to normal font.

Figure 4. Please increase the font size in the figure, then it will be well readable. 

Table 2: The chemical structures mentioned in the table2 are very confusing, please provide them separately in the bottom by referring to the table, for example using  a number or symbol.

Author Response

Dear Editors and Reviewers:

Thank you for your letter and for the Reviewers’ comments concerning our manuscript entitled “Polysaccharides from Chinese medicinal materials with the medicine and food homology: A review on their extraction, purification, structure, and biological activities  (ID: molecules-1696733). Those comments are all valuable and very helpful for revising and improving our paper, as well as the important guiding significance to our researches. We have studied comments carefully and have made correction which we hope meet with approval. Revised portions are marked in red in the paper. The main corrections in the paper and the responds to the Reviewer’s comments are as flowing:

Responds to the Reviewer’s comments:

Reviewer #1

  1. Response to comment:(In my opinion, the name "CMFHPs" is slightly misleading as research into polysaccharides from various sources has been conducted without regard to traditional Chinese medicine. Also, there is no evidence that these polysaccharides retain the same structure after the preparation of the traditional Chinese medicine using the ingredients mentioned in the review. In addition, the biological activity of various polysaccharides has been directly performed (mentioned in this review) using the extracted polysaccharides but not using traditional Chinese medicine. Most of the data presented in this review article is from plant-based polysaccharides, so it might be a good idea to change the word to medicinal plant-based polysaccharides or something similar instead of “CMFHPs”. I would also like to let you know that I am not against traditional Chinese Medicine, but it is just my suggestion. In addition, there are very few suggestions for improving the article, so I recommend publishing the article after a major revision.)

Response: First of all, thank the reviewers for their positive comments on the paper and put forward such valuable suggestions for revision. The main objective of this paper is to review the extraction, purification, structure, and biological activities of medicine and food homology polysaccharides. Medicine and food homology, and polysaccharide are the keywords in the manuscript. Ganoderma lucidum, honey, and oyster polysaccharides in medicine and food homology materials are also reviewed in the manuscript. Therefore, it is inappropriate to write medicinal plant-based polysaccharides. According to your opinion, we have changed “polysaccharides from chinese medicinal materials with medicine and food homology (CMFHPs)” to “medicine and food homology materials polysaccharides (MFHPs)” and proofread it carefully.

  1. 2. Response to comment:(Figure 1: The text in the figure 1 is not well readable, I think it is due to presented in the image format, please make the text well readable.)

Response: Thanks for the Reviewer’s suggestion. According to your opinion, we have completely revised the format and corresponding language in Figure 1.

  1. 3. Response to comment:(Figure1: In figure 1, it was written "Podwer", I think it should be "powder", please correct it.)

Response: First of all, thank you for your valuable comments. According to your opinion, we have changed " Podwer " to "Powder". In addition, we have all proofread the text in Figure 1. The specific amendments are as follows:

  1. 4. Response to comment: (Line 105: The font looks different for "Cassia seed" and is not a scientific name, please correct it to normal font.)

Response: First of all, thank you for your valuable comments. According to your opinion, we have changed "Cassia seed" to normal font. In the revised manuscript, the revised sections is marked in red to easy check.

  1. 5. Response to comment:(Figure 4. Please increase the font size in the figure, then it will be well readable.)

Response: Thanks for the Reviewer’s good evaluation and kind suggestion. According to your opinion, we have increased the font size in the Figure 4. The revised Figure 4 is easy for readers to read.

  1. 6. Response to comment: (Table 2: The chemical structures mentioned in the table2 are very confusing, please provide them separately in the bottom by referring to the table, for example using a number or symbol.)

Response: We sincerely thank you for your comments. According to your opinion, we have provided the chemical structures mentioned in Table 2 separately at the bottom of the Table 2. The specific modifications are as follows:

Note: 1. The ratios of 1→2, 1→4 in HSY-I, HSY-II, HSY-III glycosidic bonds were 5.79: 15.4: 9.39: 4.3.

  1. α-type glycosidic linkage.
  2. ICP-1 were composed of (1→), (1→4) and (1→6) glucose, (1→5) arabinose, (1→4) galac-turonic acid and (1→3,6) mannose.
  3. TPs-0 comprised a main chain ofα-Araf- (1→ 4) -α-Glcp- (1→ 3) -α-Arap- (1→ 3) -β-Galp- (1→ 3,6) -α-Galp- (1→ 5) -α-Araf- (1→ 3) -β-Galp- (1→R.
  4. GLP-1: →6)-β-D-Glcp-(1→, →6)-α-D-Galp-(1→, and→3)-β-D-Glcp-(1→ residues. GLP-2: β-D-glucan that possessing→6)-β-D-Glcp-(1→ and→3)-β-D-Glcp-(1→ residues packaged into a spherical conformation.
  5. (1→3,4)-linked-α-Rhap, (1→4)-linked-β-Galp, (1→6)-linked-β-Galp, and (1→3,6)-linked-β-Galp.
  6. α-d-Man/Glcp-(1→ and→1)-β-d-Man/Glcf-(2→ glycosidic linkageconformations.
  7. 1, 4-linked-α-D-Glcp,1, 4, 6-linked-α-D-Glcp and 1, 4-linked-α-D-Manp residues as the back bone. And the side-chains comprised of 1, 3, 5-linked-α-L-Araf, 1, 5-linked-α -L-Araf, terminalα-Araf and 1, 4-linked-β-D-Galp.
  8. AVPG-1: → 4)-α-D-Glcp-(1→ 3,4)-β-D-Glcp-(1→ 4)-α-D-Glcp-(1→. AVPG-2: → 4)-α-D-Glcp-(1→ 3,6)-β-D-Galp-(1→ 4)-α-D-Glcp-(1→.
  9. 1,3,6-linked-α-Glcp,1,4-linked-α-Glcp, 1,4-linked-β-Glcp and 1,4,6-linked-α-Glcp.
  10. 1,4-linked-β-D-Manp and 1,4-linked-β-D-Glcp with O-acetyl groups.
  11. Backbone is composed of (1,3)-linked-β-L-Rhamnose and (1,3,6)-linkage-β-D-galactose, with two branch chains of (1,4)-linked-α-D-galactose and (1,6)-linked-β-D-galactose and terminated with 1-α-L-arabinose.
  12. 1,4-linkedα-D-GalA and 1,2,4-linkedα-L-Rha. The latter was substituted at C-4 position by 1,4 linked, 1,6-linkedβ-Galp, or Teminal linkedβ-Gal.

In the revised manuscript, the revised sections is marked in red to easy check.

Special thanks to you for your good comments.

 We tried our best to improve the manuscript and made some changes in the manuscript. These changes will not influence the content and framework of the paper. Here we did not list the changes.

We appreciate for Editors/Reviewers’ warm work earnestly, and hope that the correction will meet with approval.

Once again, thank you very much for your comments and suggestions

Yours sincerely,

 Hongkun Xue PhD

Hebei University, Baoding, China

xuehk0906@hbu.edu.cn

Reviewer 2 Report

  1. The manuscript entitled “Polysaccharides from Chinese medicinal materials with the medicine and food homology: A review on their extraction, purification, structure, and biological activities” presented by Jiaqi Xuand co-authors provides a review of the extraction and purification processes, structure, biological activities, as well as potential molecular mechanisms of Chinese medicinal materials with the medicine and food homology (CMFH).

Although reasonably well prepared and within the scope of the journal, this work lacks novelty. Furthermore, this work appears to be quite similar to a previously published work (Chen, Y.; Yao, F.; Ming, K.; Wang, D.; Hu, Y.; Liu, J. Polysaccharides from Traditional Chinese Medicines: Extraction, Purification, Modification, and Biological Activity. Molecules 2016, 21, 1705. https://doi.org/10.3390/molecules21121705.)

  1. Choose a concise title for the manuscript.

  1. The manuscript's English should be checked and improved; for example, please use a comma before a coordinating conjunction (and). Check the abstract, introduction, and entire manuscript.

  1. Introduction and discussion should be focused more on the observations and novelty of this study, compared with other methods of preparation, and can be supported with related references.

  1. Figures 2,3 and 4. require a more detailed description.

Author Response

Dear Editors and Reviewers:

Thank you for your letter and for the Reviewers’ comments concerning our manuscript entitled “Polysaccharides from Chinese medicinal materials with the medicine and food homology: A review on their extraction, purification, structure, and biological activities  (ID: molecules-1696733). Those comments are all valuable and very helpful for revising and improving our paper, as well as the important guiding significance to our researches. We have studied comments carefully and have made correction which we hope meet with approval. Revised portions are marked in red in the paper. The main corrections in the paper and the responds to the Reviewer’s comments are as flowing:

Responds to the Reviewer’s comments:

Reviewer #2

  1. 1. Response to comment: [Although reasonably well prepared and within the scope of the journal, this work lacks novelty. Furthermore, this work appears to be quite similar to a previously published work (Chen, Y.; Yao, F.; Ming, K.; Wang, D.; Hu, Y.; Liu, J. Polysaccharides from Traditional Chinese Medicines: Extraction, Purification, Modification, and Biological Activity. Molecules 2016, 21, 1705. https://doi.org/10.3390/molecules21121705.)]

Response: First of all, thank you for your valuable comments. The paper you mentioned is a review of polysaccharides from traditional chinese medicines. Our paper mainly reviews the polysaccharides from medicine and food homology material. There are obvious differences between the two papers.

  1. 2. Response to comment: (Choose a concise title for the manuscript.)

Response: Thank you for your comments. According to your opinion, we have changed the title of the paper to “Polysaccharides from medicine and food homology materials: A review on their extraction, purification, structure, and biological activities”. In the revised manuscript, the revised sections is marked in red to easy check. 

  1. 3. Response to comment: (The manuscript's English should be checked and improved; for example, please use a comma before a coordinating conjunction (and). Check the abstract, introduction, and entire manuscript.)

Response: First of all, thank you for your valuable comments. According to your opinion, we have asked the experts of language service to make a comprehensive revision of the language in the paper. In the revised manuscript, the revised sections is marked in red to easy check. 

  1. 4. Response to comment: (Introduction and discussion should be focused more on the observations and novelty of this study, compared with other methods of preparation, and can be supported with related references.)

Response: First of all, we thank you for your valuable modification suggestions. In the revised manuscript, we have comprehensively revised the problems you mentioned above, and the revised sections is marked in red to easy check in the revised manuscript.  

  1. 5. Response to comment: (Figures 2,3 and 4. require a more detailed description.)

Response: Thank you for your good comments. We have revised Figures 2, 3, and 4.

Special thanks to you for your good comments.

 We tried our best to improve the manuscript and made some changes in the manuscript. These changes will not influence the content and framework of the paper. Here we did not list the changes.

We appreciate for Editors/Reviewers’ warm work earnestly, and hope that the correction will meet with approval.

Once again, thank you very much for your comments and suggestions

Yours sincerely,

 Hongkun Xue PhD

Hebei University, Baoding, China

xuehk0906@hbu.edu.cn

Reviewer 3 Report

Manuscript should be carefully checked for accuracy - This is not correct; authors wrote: In addition, polysaccharides are soluble in water but insoluble in ethanol. Starch is not soluble in water unless it is heated; cellulose is not soluble in water. This is huge problem.

Additionally:

Mounting researches – what does this mean? Authors should pay attention on selection of words

Key words should be better selected

Figure 1 – carefully check spelling (for example podwer) spaces missing … Also check other figures

In tables authors are using abbreviations which are not explained. Tables should be prepared more clearly

Author Response

Dear Editors and Reviewers:

Thank you for your letter and for the Reviewers’ comments concerning our manuscript entitled “Polysaccharides from Chinese medicinal materials with the medicine and food homology: A review on their extraction, purification, structure, and biological activities  (ID: molecules-1696733). Those comments are all valuable and very helpful for revising and improving our paper, as well as the important guiding significance to our researches. We have studied comments carefully and have made correction which we hope meet with approval. Revised portions are marked in red in the paper. The main corrections in the paper and the responds to the Reviewer’s comments are as flowing:

Responds to the Reviewer’s comments:

Reviewer #3

  1. 1. Response to comment:(Manuscript should be carefully checked for accuracy-This is not correct; authors wrote: In addition, polysaccharides are soluble in water but insoluble in ethanol. Starch is not soluble in water unless it is heated; cellulose is not soluble in water. This is huge problem.)

Response: First of all, thank you for your valuable comments. According to your suggestion, we have revised the above contents and deleted inaccurate expressions. In the revised manuscript, the revised sections is marked in red to easy check. 

  1. 2. Response to comment:(Mounting researches – what does this mean? Authors should pay attention on selection of words)

Response: Thank you for your good comments. According to your opinion, we have changed “Mounting researches” to “Accumulating evidence”. In the revised manuscript, the revised sections is marked in red to easy check.

  1. Response to comment: (Key words should be better selected)

Response: First of all, thank you for your valuable comments. According to your opinion, we have revised the keywords in the manuscript as follows: Keywords: polysaccharides; medicine and food homology; extraction and purification; structure; biological activities

  1. Response to comment: (Figure 1-carefully check spelling (for example podwer) spaces missing … Also check other figures)

Response: First of all, thank you for your valuable comments. According to your opinion, we have changed “Podwer” to “Powder”. In addition, we have all proofread the text in other figures.

  1. 5. Response to comment:(In tables authors are using abbreviations which are not explained. Tables should be prepared more clearly)

Response: Thank you for your good comments. The full names of the abbreviations in Tables 1 and 2 are given in the revised version. In the revised manuscript, the revised sections is marked in red to easy check.

Special thanks to you for your good comments.

 We tried our best to improve the manuscript and made some changes in the manuscript. These changes will not influence the content and framework of the paper. Here we did not list the changes.

We appreciate for Editors/Reviewers’ warm work earnestly, and hope that the correction will meet with approval.

Once again, thank you very much for your comments and suggestions

Yours sincerely,

 Hongkun Xue PhD

Hebei University, Baoding, China

xuehk0906@hbu.edu.cn

Round 2

Reviewer 1 Report

The article "Polysaccharides from medicine and food homology materials:A review on their extraction, purification, structure, and biological activities" has been improved compared to the previous version. The authors have also implemented my previous suggestions, hence I recommend the article to publish in the present form. 

Reviewer 2 Report

I am satisfied with the author's responses to my questions

Reviewer 3 Report

Even though authors did make some changes, I dont think that this review can be accepted.